# Modulation of Macrophage Activity by Pulsed Electromagnetic Fields in the Context of Fracture Healing

**DOI:** 10.3390/bioengineering8110167

**Published:** 2021-10-29

**Authors:** Yangmengfan Chen, Maximilian M. Menger, Benedikt J. Braun, Sara Schweizer, Caren Linnemann, Karsten Falldorf, Michael Ronniger, Hongbo Wang, Tina Histing, Andreas K. Nussler, Sabrina Ehnert

**Affiliations:** 1Siegfried Weller Research Institute, BG Trauma Center Tübingen, Department of Trauma and Reconstructive Surgery, University of Tübingen, Schnarrenbergstr. 95, D-72076 Tübingen, Germany; chenyangmengfantj@163.com (Y.C.); mmenger@bgu-tuebingen.de (M.M.M.); bbraun@bgu-tuebingen.de (B.J.B.); sara.schweizer80@googlemail.com (S.S.); caren.linnemann@student.uni-tuebingen.de (C.L.); thisting@bgu-tuebingen.de (T.H.); andreas.nuessler@gmail.com (A.K.N.); 2Sachtleben GmbH, Haus Spectrum am UKE, Martinistraße 64, D-20251 Hamburg, Germany; falldorf@citresearch.de (K.F.); ronniger@citresearch.de (M.R.); 3Union Hospital, Tongji Medical College, Huazhong University of Science and Technology, 1277 Jiefang Ave., Wuhan 430022, China; whbdf@yahoo.com

**Keywords:** extremely low frequency pulsed electromagnetic fields (ELF-PEMFs), macrophages, mesenchymal stem/stromal cells, extracellular matrix, fracture healing

## Abstract

Delayed fracture healing and fracture non-unions impose an enormous burden on individuals and society. Successful healing requires tight communication between immune cells and bone cells. Macrophages can be found in all healing phases. Due to their high plasticity and long life span, they represent good target cells for modulation. In the past, extremely low frequency pulsed electromagnet fields (ELF-PEMFs) have been shown to exert cell-specific effects depending on the field conditions. Thus, the aim was to identify the specific ELF-PEMFs able to modulate macrophage activity to indirectly promote mesenchymal stem/stromal cell (SCP-1 cells) function. After a blinded screening of 22 different ELF-PEMF, two fields (termed A and B) were further characterized as they diversely affected macrophage function. These two fields have similar fundamental frequencies (51.8 Hz and 52.3 Hz) but are emitted in different groups of pulses or rather send–pause intervals. Macrophages exposed to field A showed a pro-inflammatory function, represented by increased levels of phospho-Stat1 and CD86, the accumulation of ROS, and increased secretion of pro-inflammatory cytokines. In contrast, macrophages exposed to field B showed anti-inflammatory and pro-healing functions, represented by increased levels of Arginase I, increased secretion of anti-inflammatory cytokines, and growth factors are known to induce healing processes. The conditioned medium from macrophages exposed to both ELF-PEMFs favored the migration of SCP-1 cells, but the effect was stronger for field B. Furthermore, the conditioned medium from macrophages exposed to field B, but not to field A, stimulated the expression of extracellular matrix genes in SCP-1 cells, i.e., *COL1A1*, *FN1*, and *BGN*. In summary, our data show that specific ELF-PEMFs may affect immune cell function. Thus, knowing the specific ELF-PEMFs conditions and the underlying mechanisms bears great potential as an adjuvant treatment to modulate immune responses during pathologies, e.g., fracture healing.

## 1. Introduction

Almost every human on earth experiences one or more fractures during life. Although the treatment options for fractures have been greatly developed, 5% to 10% of all fractures still result in delayed healing or even non-union [1]. Consequently, fractures are a major cause of disability, morbidity, and even mortality, particularly in elderly patients leading to a high economic burden [2].

Fracture healing is a complex and dynamic process that can be divided into three partially overlapping phases: inflammation phase, reparative phase, and remodeling phase. The complex and excellently tuned process during fracture healing requires the involvement of various cell types [3]. Neutrophils, arriving within a few hours at the fracture site, represent the first line of defense. They play a crucial role in resolving the formed hematoma and initiating inflammatory responses. Monocytes/macrophages represent a large proportion of the immune cells present at the injury site throughout the entire healing phase; however, their activation status is changing [4]. After a fracture occurs, monocytes/macrophages are recruited to the injury site, where they differentiate into macrophages within 2 to 3 days [5]. Unlike monocytes, macrophages show high plasticity. After receiving adequate stimuli, macrophages may easily change their activity and function, exhibiting either pro-inflammatory or anti-inflammatory and pro-healing activities [6,7]. Macrophage function is largely mediated by factors such as cytokines, growth factors, and chemokines, which they secrete depending on their activation status. Thus, macrophages are involved in controlling the different phases of the healing process by adapting their phenotype [8]. The differentially activated macrophages are mutually exclusive and involved in the whole healing process [9]. In the acute inflammatory phase, pro-inflammatory macrophages are important. These cells infiltrate the site of injury not only to detect and remove pathogenic microbes and cellular debris but also to initiate various signaling pathways required for the subsequent repair phase. In the chronic inflammatory and reparative phase, the tissue-resident macrophages exhibit anti-inflammatory and pro-healing activities to reduce hyper-inflammation and promote tissue regeneration. Later, during tissue remodeling, a controlled amount of pro-inflammatory macrophages are required [4]. Therefore, regulating the immune microenvironment at the site of injury by modulating macrophage activity is crucial for fracture healing [10].

Disorders in macrophage activity are highly related to the failure of fracture healing. For example, unrestrained and persistent pro-inflammatory macrophages may lead to impaired fracture healing, particularly in patients with rheumatoid arthritis (RA), chronic obstructive pulmonary disease (COPD), diabetes mellitus (DM), liver fibrosis/cirrhosis, and systemic lupus erythematosus (SLE) [11]. However, the suppression of inflammation also arrests the process of fracture healing [12,13]. Therefore, neither a hyper-inflammatory state nor an immune-suppressive state alone can contribute to successful fracture healing. The key factor to successful fracture healing is a dynamic and suitable immune microenvironment at the injury site. Thus, the modulation of inflammatory responses may offer great therapeutic options to support the healing of fractures and soft tissues.

Exposure to extremely low frequency pulsed electromagnetic fields (ELF-PEMF) is a non-invasive, penetrable, and patient-friendly treatment. Therefore, ELF-PEMFs are a promising adjuvant treatment to dynamically regulate the local immune microenvironment and promote healing processes. In recent years, ELF-PEMF was proven to effectively support fracture healing and bone regeneration (overview see [14]). Positive effects to the proposed mechanisms on osteoblasts [15], mesenchymal stem/stromal cells (MSCs) [16], chondrocytes [17], and intervertebral disc cell [18] function have been reported. However, there has been little research focused on the immune-regulatory ability of ELF-PEMFs [19], even though the importance of osteoimmunology is generally acknowledged. The reported effects of the ELF-PEMFs on macrophages are diverse. When employing different readout parameters, some reports show pro-inflammatory effects [20,21,22], while others reported anti-inflammatory [23], or even immune-suppressive [24], effects of the ELF-PEMFs. Interestingly, the ELF-PEMFs in these four studies all had a fundamental frequency of 50 Hz but varied in their magnetic field density and pulse pattern.

Earlier reports suggested that each cell type responds to ELF-PEMFs with specific fundamental frequencies and pulse burst patterns. While an ELF-PEMF with a fundamental frequency of 16 Hz was shown to improve the function of osteoblasts [15,25], another ELF-PEMF (comparable intensity) with a fundamental frequency of 26 Hz also affected osteoclast function [16]. Studies on primary rat calvaria cells showed that not only the fundamental frequency but also the waveform and pulse burst pattern might be relevant [26,27]. Thus, in an initial blinded screening, the effect of 22 ELF-PEMFs, with different fundamental frequencies, waveforms and pulse burst patterns on macrophage activation, were tested. Among these, two ELF-PEMF were chosen to further characterize their distinct effect on macrophage activity and function. This includes direct effects on macrophages: (i) expression of phenotypic markers; (ii) formation of reactive oxygen species (ROS) as an inflammatory response; and (iii) secretion of cytokines, growth factors and chemokines, as well as the expected effects that the treated macrophages have on MSCs during fracture healing: (i) migration and (ii) formation of extracellular matrix (ECM) components.

## 2. Materials and Methods

If not specified, reagents, culture media, and medium supplements were purchased from Merck (Darmstadt, Germany).

### 2.1. Human Material

All experiments involving human materials strictly adhered to the Declaration of Helsinki (1964) in its latest amendment and were approved by the Ethics Committee of the University Clinic Tübingen (541/2016BO2 approved 09.08.2016). PBMCs were isolated from the venous blood of healthy volunteers. Blood was obtained with the signed informed consent of the donors.

### 2.2. Isolation of Peripheral Blood Mononuclear Cells (PBMCs)

PBMCs were isolated from fresh EDTA blood. Venous blood was collected in EDTA tubes (S-Monovette, Sarstedt, Sarstedt, Germany) and directly used for density gradient centrifugation. A total of 6 mL of blood was carefully layered on 6 mL of Lympholyte-poly Cell Separation Medium (Cedarlane, Burlington, ON, Canada). Samples were centrifuged for 35 min at 500 g without a break at room temperature. The PBMC (upper) layer was transferred to a new tube. After washing twice with PBS, the cells were counted and seeded at a concentration of 5 × 10^5^ cells/mL in an RPMI 1640 medium with 2% autologous plasma [28]. Experiments were performed at 37 °C (5% CO_2_, humidified atmosphere).

### 2.3. ELF-PEMF Device and Exposure

The ELF-PEMF devices (Somagen^®^, Sachtleben GmbH, Hamburg, Germany) are medical devices certified according to European law (CE 0482, compliant with EN ISO 13485:2016 + EN ISO 14971:2012). The device generates an AC magnetic field, here ELF-PEMF, via applicators (coils). Furthermore, the applicators distort the local earth magnetic DC-field, yielding inhomogeneous DC-field conditions [25]. In this study, 22 ELF-PEMF conditions have been tested in a blinded manner. The 22 ELF-PEMF all have a similar intensity (with a magnetic field amplitude between 6 and 282 µT at 6 mm above the applicator) but different fundamental frequencies, emitted in different pulse burst patterns (pulses in send–pause intervals) [14]. The two ELF-PEMFs identified to modulate macrophage function have fundamental frequencies close to each other (field A: 51.8 Hz and field B: 52.3 Hz) but differ in their pulse burst pattern. The daily ELF-PEMF exposure was 7 or 30 min. Unblinding of the ELF-PEMF conditions was done after all experiments were finished and evaluated.

### 2.4. Western Blot

PBMCs were lysed in an ice-cold RIPA buffer (50 mM TRIS, 250 mM NaCl, 2% NP40, 2.5 mM EDTA, 0.1% SDS, 0.5% DOC, and protease/phosphatase inhibitors: 1 μg/mL Pepstatin, 5 μg/mL Leupeptin, 1 mM PMSF, 5 mM NaF, and 1 mM Na_3_VO_4_). The lysate was centrifuged (14,000× *g*, 10 min) to remove cell debris. Protein concentration was determined by micro-Lowry; 25 μg of total protein were separated by SDS-PAGE (10% acrylamide-bisacrylamide gels, 100 V, 180 min) then transferred to nitrocellulose membranes (100 mA, 180 min). Ponceau staining was used to confirm protein separation and transfer; 5% BSA was used to block unspecific binding sites. Then, membranes were incubated with primary antibodies against CD86, Arginase 1 (sc-28347, sc-20150 from Santa Cruz Biotechnology, Heidelberg, Germany), phospho-Stat1 (7649 from Cell Signaling Technologies, Danvers, MA, USA), and GAPDH (G9545 from Sigma-Aldrich, Munich, Germany) diluted in TBST, overnight at 4 °C. The following day, the membranes were incubated with the corresponding HRP-labeled secondary antibodies (1:10,000 in TBST) for 2 h at room temperature. After washing, target proteins were visualized with an enhanced chemiluminescent (ECL) substrate solution (1.25 mM Luminol, 0.2 mM p-Coumaric acid, 0.03% H_2_O_2_ in 100 mM TRIS, pH = 8.5), and the chemiluminescent signals were detected with a CCD camera. Signal intensities were quantified using ImageJ software [25].

### 2.5. DCFH-DA Assay

Oxidative stress was detected using a 2′, 7′-dichlorofluorescein diacetate (DCFH-DA) assay, detecting different reactive oxygen species. Freshly isolated PBMCs were incubated with 10 µM DCFH-DA for 25 min at 37 °C. Cells were washed once with PBS. Then cells were exposed to the ELF-PEMF, and positive control cells were stimulated with 0.001% H_2_O_2_. For a time course of 20 min, the increase in fluorescence (ex/em = 485/520 nm) was detected with the omega microplate reader, the slope representing an accumulation of O_2_^−^, H_2_O_2_, HO and ONOO^−^ [29].

### 2.6. Human Cytokine Array C5 with Media Conditioned by PBMC Exposed to ELF-PEMFs

The human Cytokine Array C5 (RayBiotech, Peachtree Corners, GA, USA) was used to characterize media conditioned by PBMCs. Briefly, PBMCs (5 × 10^5^ cells/mL) were exposed to the different ELF-PEMF then cultured for 24 h at 37 °C (5% CO_2_, humidified atmosphere). Cells were removed from the conditioned media by centrifugation (1000× *g*, 10 min). The array was performed according to the manufacturer’s instructions. Chemiluminescent signals were detected with the ChemoCam and quantified using ImageJ software. The data were normalized to the internal controls [28].

### 2.7. Culture and Differentiation of SCP-1 Cells

This study used the human immortalized mesenchymal stem cell line SCP-1, kindly provided by Professor Matthias Schieker, as an osteogenic precursor cell. The SCP-1 cells were cultured in α-MEM medium (Gibco, Darmstadt, Germany) supplemented with 5% fetal bovine serum (FBS) in a water-saturated atmosphere of 5% CO_2_ at 37 °C. The osteogenic function was induced with a differentiation medium (α-MEM medium supplemented with 1% FBS, 200 μM L-ascorbate-2-phosphate, 5 mM β-glycerol-phosphate, 25 mM HEPES, 1.5 mM CaCl_2_, and 100 nM dexamethasone) mixed 1:1 with a conditioned medium from ELF-PEMF exposed PBMCs. The SCP-1 cells themselves were not exposed to the ELF-PEMF.

### 2.8. Cell Migration Assay

Cell migration was evaluated using the cell migration assay kit (tebu-bio GmbH, Offenbach, Germany). Sterilized stoppers were placed in 96-well plates before seeding the SCP-1 cells at a concentration of 4 × 10^5^ cells/mL. After 24 h, the stoppers were removed from the wells, and the cells were washed 3 times with PBS. Then, the growth medium and conditioned medium from ELF-PEMF exposed PBMCs were added in a 1:1 ratio. Immediately, an image was taken with the microscope to document the time point 0 h. After 48 h, SRB staining was performed for better visualization of SCP-1 cells. The “gap closure” in the microscopic images was calculated and analyzed using ImageJ software [28].

### 2.9. Sulforhodamine B (SRB) Staining

Adherent cells were fixed with ice-cold 99% EtOH for at least 1 h at −20 °C. After washing the plates with tap water, the cells were covered with SRB solution (0.4% SRB in 1% acetic acid) for 30 min. The unbound SRB was removed by washing with 1% acetic acid [28].

### 2.10. RNA Isolation and RT-PCR

Total mRNA was isolated by phenol–chloroform extraction. The obtained mRNA was dissolved in DEPC water. The total mRNA content was determined photometrically (λ = 260 nm, 280 nm, and 320 nm) with the omega microplate reader, and mRNA integrity was confirmed using agarose gel electrophoresis. The total mRNA was converted into cDNA using the First Strand cDNA Synthesis Kit (Thermo Fisher Scientific, Sindelfingen, Germany) according to the manufacturers’ instructions. RT-PCR was carried out using the 2× Red Taq Mastermix (Biozym, Oldendorf, Germany) [25]. Optimized PCR conditions for each primer set are given in Table 1. Primers were designed with the help of primer-BLAST with the respective gene bank accession number listed in the table.

### 2.11. Statistical Analysis

Results are presented as box plots (min. to max.) with individual data points. Each experiment was repeated at least three times (N ≥ 3) with a minimum of three independent replicates (*n* ≥ 3). The exact number of biological (N) and technical replicates (*n*) for each experiment is given in the figure legends. Statistical analyses were performed using the GraphPad Prism software version 8. Data sets were compared using a non-parametric Kruskal–Wallis test, followed by Dunn’s multiple comparison test. A *p*-value below 0.05 was considered statistically significant.

## 3. Results

### 3.1. ELF-PEMFs Exposure Modulates Macrophage Differentiation

In this study, the effect of 22 different ELF-PEMFs on macrophage activity was screened in a blinded manner in a two-step procedure. The ELF-PEMFs had comparable intensities but different fundamental frequencies ranging from 3.3 Hz to 90.60 Hz, emitted in pulses or bursts in send–pause intervals (pulse burst pattern). In the first screening step, freshly isolated PBMCs were activated with 200 nM PMA and directly exposed to the different ELF-PEMFs for 7 min each. After 24 h, activation of macrophages was judged by attaching the cells to culture plastic. Furthermore, a possible effect of the macrophage conditioned medium on SCP-1 cell migration was considered (Figure 1). The number of ELF-PEMFs was reduced to four in a second screening step, and a second duration of exposure (30 min) was added for each. Of the four ELF-PEMF (blinded), two ELF-PEMFs (termed A and B) showed opposing effects on macrophage function and were therefore further investigated for unblinding.

In the screening, two ELF-PEMF were included, which showed positive effects on osteoblast function [15,25,29] and osteoclast function in earlier studies [16]. These two ELF-PEMF, however, did not seem to affect macrophages in our setting.

As previously described, freshly isolated PBMCs were activated with 200 nM PMA and directly exposed to the two ELF-PEMFs for 7 min or 30 min each. After 24 h, cells were lyzed, and markers of pro-and anti-inflammatory macrophages were detected by Western blot (Figure 2). Exposure to field A significantly increased the protein levels of phosphorylated Stat1 (2.9-fold with *p* = 0.0084/2.8-fold with *p* = 0.0396) and CD86 (2.0-fold with *p* = 0.0013/1.6-fold with *p* = 0.0193), which are markers for pro-inflammatory macrophages (Figure 2B,C). Inversely, exposure to field B significantly increased (2.6-fold with *p* = 0.0027/2.5-fold with *p* = 0.0034) the protein levels of Arginase I, a marker for anti-inflammatory macrophages, known to support healing processes (Figure 2D). Interestingly, extending the exposure time from 7 min to 30 min could not increase the observed effect.

### 3.2. Exposure to Field A Leads to the Formation of Intracellular Reactive Oxygen Species (ROS)

As a marker of inflammation, intracellular ROS levels were detected. It can already be determined that 7 min exposure to field A, but not exposure to field B, significantly increased the amount of intracellular ROS (1.4-fold with *p* = 0.0004) in freshly isolated PBMCs activated with 200 nM PMA (Figure 3).

### 3.3. Field A and Field B Diversely Regulate the Secretion of Cytokines, Growth Factors, and Chemokines by Macrophages

The paracrine effect of macrophages on neighboring cells, e.g., mesenchymal stem/stromal cells (MSCs) and other immune cells in the fracture site, is essential for the healing outcome [4]. Freshly isolated PBMCs were activated with 200 nM PMA and directly exposed to the two ELF-PEMFs for 7 min or 30 min each. After 24 h, cells in the conditioned medium were collected, and secreted factors were detected with the human Cytokine Array C5 (RayBiotech, Peachtree Corners, GA, USA). The resulting heat maps show that exposure to field A and field B diversely regulated the secretion of cytokines, growth factors, and chemokines in the macrophages. Many important pro-inflammatory cytokines, e.g., interleukin-1 beta (IL-1β), IL-3, interferon-gamma (IFN-γ), or Oncostatin M (OSM), were elevated in a conditioned medium from macrophages that were exposed to field A (Figure 4A). Inversely, many anti-inflammatory cytokines, e.g., IL-10, transforming growth factor-beta (TGF-β) isoforms, insulin-like growth factor (IGF), and their binding proteins (IGFBPs), were elevated in conditioned medium from macrophages that were exposed to field B (Figure 4A,B). Including angiogenin, these factors secreted by macrophages after exposure to field B have been reported to enhance regeneration during tissue repair [3]. Similar to the cytokines and growth factors, secretion of chemokines was selectively induced upon exposure to filed A and B (Figure 4C), which may affect cell invasion into the injury site. Interestingly, prolonging the exposure time from 7 min to 30 min seemed to intensify the observed effect.

### 3.4. Factors Secreted by Macrophages Exposed to the Two ELF-PEMF Stimulated Migration of SCP-1 Cells

To investigate how the ELF-PEMF induced alterations in the macrophages cytokine secretion affect migration of MSCs, a migration assay with SCP-1 cells (immortalized human bone marrow-derived MSCs [30]) was performed (Figure 5). For the entire migration phase (48 h), a macrophage conditioned medium (1:1 ratio) was added to the SCP-1 cells. Cells invading into the migration zone were visualized with SRB staining, and “gap closure” was determined with the help of ImageJ software. The quantitative results indicated that conditioned medium from macrophages exposed for 7 min to field B already promoted the migration of SCP-1 cells (1.7-fold with *p* = 0.0390). In line with the data from the human Cytokine Array *C5* conditioned medium from macrophages exposed for 30 min to field B, had even stronger effects on the migration of SCP-1 cells (1.7-fold with *p* = 0.0050). In contrast, the presence of conditioned medium from macrophages exposed to field A failed to significantly promote migration of SCP-1 cells (1.5-fold with *p* = 0.0731 and 1.5-fold with *p* = 0.0902).

### 3.5. Conditioned Medium from Macrophages Exposed to Field B Induced Extracellular Matrix Formation in SCP-1 Cells

Directly upon invasion into the tissue, MSCs start producing extracellular matrix components. A well-organized matrix formation at an early stage decides the outcome of tissue regeneration. To investigate this aspect, SCP-1 cells were cultured in the presence of the different macrophage conditioned media for 1 week. Then, the gene expression of matrix proteins *collagen 1A1*, *fibronectin*, *biglycan*, and *versican*, was evaluated (Figure 6A). Results showed that the conditioned medium from macrophages exposed to field A did not affect the expression of *collagen 1A1*, *fibronectin*, *biglycan*, and *versican*. In contrast, the conditioned medium from macrophages exposed to field B induced expression of *collagen 1A1* (2.2-fold with *p* < 0.0001), *fibronectin* (1.3-fold with *p* = 0.0106), and *biglycan* (1.7-fold with *p* = 0.0004), however, mostly with the shorter ELF-PEMF exposure (7 min not 30 min). Similarly, the expression of *versican* was not affected in SCP-1 cells cultured in the presence of a conditioned medium from macrophages exposed to field B (Figure 6B–D).

## 4. Discussion

Bone marrow is the major reservoir for immune cells in the human body, playing a crucial role in the innate immune response [31]. Locally, bone cells and immune cells tightly communicate and cooperate to form an ‘osteoimmune micro-environment’ [32]. Immediately after a bone is fractured, disruption of blood vessels leads to the formation of a hematoma at the injury site. Within hours to days, a large variety of immune cells, e.g., neutrophils, monocytes, macrophages, T-cells, and others, invade the hematoma [33]. Besides the immediate pathogen defense, immune cells orchestrate the following healing process mainly by secreting cytokines, growth factors and chemokines. Macrophages are present throughout the entire healing process and can change their activation status upon demand [3,4]. The high plasticity of the macrophages [9], in combination with their long life span at the fracture site [34], identifies macrophages as possible target cells when attempting to temporally modulate the osteoimmune micro-environment for therapeutic purposes, i.e., when exploring the effects of different ELF-PEMFs.

In our blinded screening, we identified two ELF-PEMF that have diversely affected macrophage function. Unblinding revealed that the fundamental frequencies of these two ELF-PEMF are very close to each other, namely 51.8 Hz for field A and 52.3 Hz for field B. In literature, another 50 Hz ELF-PEMF induced anti-oxidative defense mechanism in LPS challenged THP-1 cells (mononuclear), thus initiating an anti-inflammatory reaction often associated with the M2 phenotype [23]. Yet other 50 Hz ELF-PEMFs induced ROS and thus pro-inflammatory responses in mouse macrophages [20,21], a functional response associated with the M1 phenotype. Similarly, another 50 Hz ELF-PEMF affected the immune response of monocyte-derived macrophages towards pathogens by modulating, amongst others, intracellular NO [24]. These findings are all in line with our initial assumption that different cell types respond to a specific range of fundamental frequencies. However, this does not explain why the two fields have opposing effects on the macrophages. In one study, the magnetic field density of the 50 Hz ELF-PEMF was altered (0.5–1.5 mT), which had an effect on the phagocytic activity of the macrophages [21]. This is in line with another report, which showed increased phagocytic activity and IL-1β release in mouse macrophages exposed to 50 Hz ELF-PEMF (1 mT) for 45 min [22]. In the latter study, ELF-PEMF exposure also induced ROS formation in the macrophages; however, ROS formation was independent of the chosen magnetic field density (0.05–1 mT) [22]. In our experiments, the produced ELF-PEMFs were inhomogeneous over the Somagen^®^ applicator; the intensities of our two fields were similar. Their major difference between field A and field B was the pattern in which the pulses and bursts were emitted. The so-called send–pause intervals, suggesting that this could be a crucial factor in our experiments.

The screening also included ELF-PEMFs that have shown effects on osteoblasts (16 Hz [15,25]) and osteoclasts (26 Hz [16]) in previous studies. These two ELF-PEMFs did not affect macrophages in our screening, further underlining the idea of a frequency “window” for different cell types [35]. For example, the 16 Hz ELF-PEMF induced osteoblast function in patients undergoing high tibia osteotomies. However, no effect on osteoclast inflammatory markers was observed [36]. By changing the frequency “window”, other cells might be addressed. This offers the possibility to target specific cells in different pathologies. With regard to macrophages, suppress or induce inflammation when required. An example is the two ELF-PEMF investigated here: CD86 and phospho-Stat1 are two common markers for pro-inflammatory macrophages [37]. Our data show an increase of CD86 and phospho-Stat1 after exposure of PMA-stimulated PBMCs to field A but not after exposure to field B. Arginase I, a common marker for anti-inflammatory macrophages, in turn, was increased after exposure of PMA-stimulated PBMCs to field B, but not after exposure to field A. These effects were not enhanced by prolonging the duration of the exposure. These data indicated that exposure to field A triggers macrophages towards a pro-inflammatory phenotype. In contrast, the exposure to field B triggers macrophages towards an anti-inflammatory and pro-healing phenotype [37].

To further investigate the proposed contrary effects of the two ELF-PEMF on macrophages, ROS production was determined. An accumulation of ROS is frequently observed in pro-inflammatory macrophages and is directly associated with the cells’ phagocytic activity [38,39]. In line with the previous observation, only exposure to field A, but not field B, led to an increased accumulation of ROS in the ELF-PEMF exposed macrophages. The diverse effects of our two ELF-PEMF fields clearly showed that ELF-PEMF conditions are critical for the cell-specific effect. In macrophages, ROS are mainly formed by NADPH oxidases (NOX) [39], but also as a downstream product of the TLR signaling pathway [40]. In macrophages, NOX2-dependent mitochondrial ROS have direct antimicrobial activity, while ROS generated by several other NOX enzymes are supposedly involved in combating infections with protozoan parasites. Whether ROS regulate inflammatory responses of macrophages seems to be dependent on factors, e.g., type and cellular localization of the ROS and the respective stimuli [41,42]. It was described that stimulation via TLRs leads to an increase in mitochondria-derived ROS (mtROS), mainly O_2_, produced by complex I and III and its downstream product H_2_O_2_. Upon release into the cytoplasm, mtROS caused the dimerization of the NFκB essential modulator NEMO, subsequent activation of NFκB signaling and increased production of inflammatory cytokines [43]. Other reports show that cytoplasmically released mtROS may disrupt the interaction of the thioredoxin-interacting protein with thioredoxin, favoring the assembly of the NRLP3 inflammasome and subsequent activation of caspase-1 and related release of IL-1β and IL-18 [44]. These pro-inflammatory effects of mtROS are thought to be independent of NOX2 activity [45]. In contrast, following phagocytosis, NOX2 is activated to produce phagosomal ROS. By inactivating cathepsins L and S, phagosomal ROS inhibits excessive proteolysis of engulfed proteins, thus supporting the presentation of antigens by major histocompatibility complex class II molecules. These examples show that a tightly controlled increase in cellular ROS, a condition termed oxidative eustress [46], is an important regulator for immunological processes [47].

To verify the immune-regulatory ability of ELF-PEMFs, we performed the human Cytokine Array C5 to identify cytokines, growth factors, and chemokines secreted by the ELF-PEMF exposed macrophages. Compared to unstimulated (no ELF-PEMF) macrophages, exposure to field A stimulated the secretion of pro-inflammatory cytokines and exposure to field B stimulated the secretion of anti-inflammatory cytokines and growth factors known to support healing processes. In contrast to the phenotypic markers, elongation of the exposure duration from 7 min to 30 min enhanced the observed effect. For example, secretion of IL-10, a factor promoting the formation of a bone matrix [48], strongly increased only when macrophages were exposed for 30 min to field B. The same holds for the TIMP1 and TIMP2–secretion of both tissue inhibitors for matrix metalloproteinases (MMPs) was strongly upregulated when macrophages were exposed for 30 min to field B. This is of special interest, as patients with delayed fracture healing and non-union frequently show elevated activity of MMPs and decreased levels of TIMPs [49]. However, timing has to be considered when considering to increase secretion of TIMPs by exposure to field B—prolonged inhibition of MMPs, especially MMP9, was reported to result in defective endochondral ossification, diminished ECM remodeling, and delayed vascularization during skeletal healing [50]. This thought is fostered by the results on osteoprotegerin. The soluble antagonist for receptor activator of NF-κB ligand suppresses osteoclastogenesis, which is critically required in the bone remodeling phase [51]. Interestingly, the two ELF-PEMF induce the secretion of different angiogenic factors in macrophages. Exposure to the more anti-inflammatory field B induced factors, e.g., angiogenin, which besides angiogenesis, is also involved in various physiological and pathological processes through regulating cell proliferation, survival, migration, invasion, and/or differentiation [52]. In contrast, exposure to the more pro-inflammatory field A stimulated macrophages to secrete factors, e.g., vascular endothelial growth factor (VEGF). Here, the timing of the ELF-PEMF exposure seems to be critical when considering to increase secretion of VEGF by exposure to field A–while a rapid (few days) increase in VEGF serum levels is desired directly after a fracture. Prolonged elevation in VEGF serum levels is associated with delayed fracture healing and fracture non-unions [53]. A similar curve to VEGF was reported for transforming growth factor-beta (TGF-β). A rapid increase after fracture is required for successful fracture healing, but also its rapid decline after a few days [54]. There are several pathologies that regulate TGF-β levels–chronically elevated TGF-β levels are found in patients with chronic inflammation, e.g., patients with diabetes mellitus, COPD, liver- or kidney- fibrosis/cirrhosis, frequently displaying secondary osteoporosis with increased fracture risk and delayed fracture healing. One possible reason might be a reduced sensitivity of the bone cells towards mechanical stimulation [55]. Reduced TGF-β levels are found, for example, in smokers, which is associated with poor fracture healing [56]. Exposure to field B strongly increased the secretion of all three TGF-β isoforms. Thus, it is feasible to use exposure to field B not only considering the timing of the ELF-PEMF exposure after the fracture but also the patients’ current condition and medical history.

Besides its immune-modulatory function, TGF-β also acts as a chemokine and induces the expression of ECM proteins [57]. Thus, it was not surprising that the secretome of macrophages exposed to field B better stimulated migration of SCP-1 cells than the secretome of macrophages exposed to field A. Secretome from macrophages with the prolonged (30 min) ELF-PEMF exposure was more potent to induce SCP-1 cell migration than secretome from macrophages with the shorter (7 min) ELF-PEMF exposure. This can be explained by the results from the human Cytokine Array C5, showing enhanced secretion of cytokines, chemokines and growth factors when the duration of the ELF-PEMF exposure was prolonged.

When arriving at the fracture site, MSC ideally induces ECM synthesis, maturation, and subsequent mineral deposition [58]. In the correct composition, the formed ECM then provides a suitable niche for MSCs, regulates several intracellular signaling pathways, and thus controls the proliferation and maturation of MSCs [59]. As expected from the increased TGF-β content in the secretome of macrophages exposed to field B, the expression of collagen 1A1, fibronectin and biglycan was increased in SCP-1 cells. Expression of versican showed an inverse trend, being downregulated in SCP-1 cells cultured with secretome of macrophages exposed to field A. These results also indicate that the shorter 7 min exposure to field B was sufficient to produce enough cytokines and growth factors to exert a positive effect on ECM production in SCP-1 cells.

## 5. Conclusions

In summary, this study demonstrated that ELF-PEMFs with specific parameters might act immune-regulatory. Using these specific ELF-PEMFs as an adjuvant treatment to modulate the osteoimmune micro-environment at a fracture site is promising to promote fracture healing. However, not only the individual history and thus needs of the patients have to be considered, but also the duration and timing of the treatment have to be critically controlled. An ability to online measure the relevant factors, e.g., by suitable sensors and analytics, and then modify the ELF-PEMF conditions accordingly could be a future perspective to individualize care and accelerate healing, requires further research.

## Figures and Tables

**Figure 1 bioengineering-08-00167-f001:**
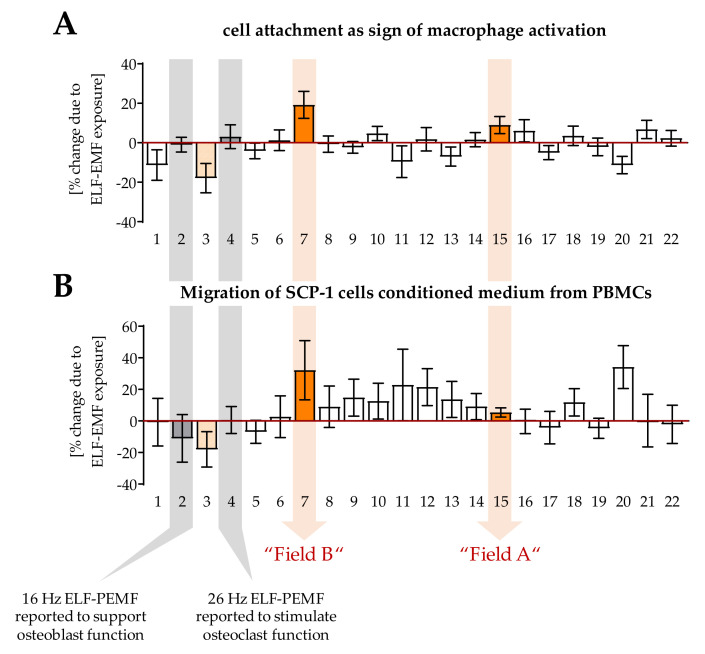
Screening of 22 different ELF-PEMF on macrophage function. Freshly isolated PBMCs (N = 7) were stimulated with 200 nM PMA and exposed to the different ELF-PEMFs for 7 min each in a blinded manner. (**A**) After 24 h, cell attachment was determined by Hoechst 33342 staining, and (**B**) conditioned medium was collected and added (1:1) to SCP-1 cells in a migration assay. The ELF-PEMF effects are displayed as fold of control (without ELF-PEMF exposure). After unblinding, field #2 was identified as the ELF-PEMF, which showed positive effects on osteoblast function in earlier studies [15,25,29] and field #4 was identified as the ELF-PEMF, which additionally stimulated osteoclast function in earlier studies [16]. Fields #3, #7, #15, and #20 were further investigated, of which #7 and #15 showed diverse effects on macrophage function—termed field A and field B in the following experiment.

**Figure 2 bioengineering-08-00167-f002:**
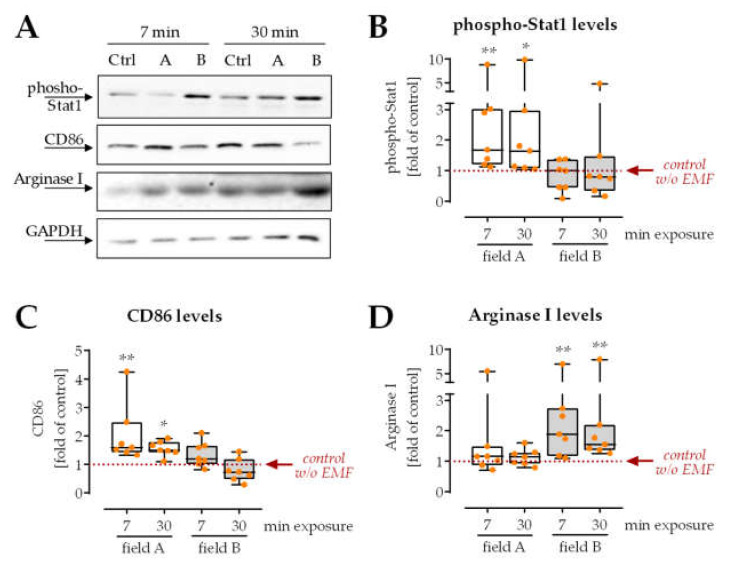
Macrophage differentiation is altered after single exposure to ELF-PEMFs. Freshly isolated PBMCs were exposed to the two ELF-PEMFs for 7 min or 30 min, respectively. After 24 h, markers of pro-and anti-inflammatory macrophages were detected by Western blot. (**A**) Representative Western blot images. All uncropped Western blot images are shown in Appendix A. (**B**,**C**) As markers of pro-inflammatory macrophages, protein levels of phosphorylated Stat1 and CD86 were determined. (**D**) As markers of anti-inflammatory and pro-healing macrophages, protein levels of Arginase I were determined. N = 7, *n* = 3. Data were compared by non-parametric Kruskal–Wallis test, followed by Dunn’s multiple comparison test: * *p* < 0.05, and ** *p* < 0.01 as compared to control cells without ELF-PEMF exposure.

**Figure 3 bioengineering-08-00167-f003:**
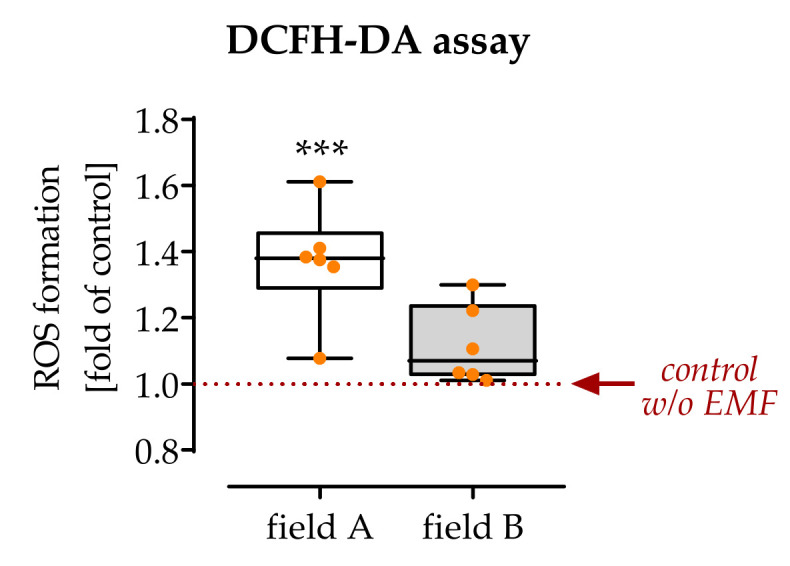
Intracellular ROS levels were regulated after single exposure to ELF-PEMFs. To detect intracellular reactive oxygen species (ROS), cells were incubated with a DCFH-DA probe before exposure to the ELF-PEMFs for 7 min. Produced ROS was quantified by the green fluorescence formed. N = 6, *n* = 3. Data were compared by non-parametric Kruskal–Wallis test, followed by Dunn’s multiple comparison test: *** *p* < 0.001 as compared to control cells without ELF-PEMF exposure.

**Figure 4 bioengineering-08-00167-f004:**
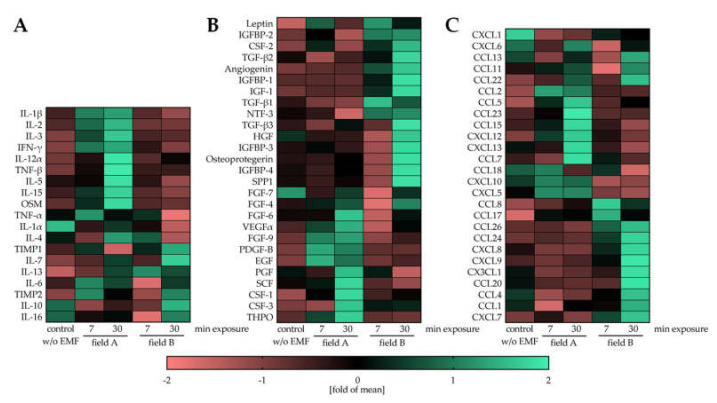
Cytokines, growth factors and chemokines secreted by PBMCs after exposure to the ELF-PEMFs. Freshly isolated PBMCs (N = 5) were stimulated with 200 nM PMA and exposed to the two ELF-PEMF for 7 min or 30 min, respectively. After 24 h, the conditioned medium was collected, pooled and analyzed (*n* = 3) for secreted factors using the human Cytokine Array C5 (RayBiotech, Peachtree Corners, GA, USA). Data were normalized with the standard score (z-score) method. Data are presented as heat maps: (**A**) Heat map of cytokines related to immune function and inflammation; (**B**) Heat map of cytokines and growth factors involved in repair and healing processes; and (**C**) Heat map of chemokines.

**Figure 5 bioengineering-08-00167-f005:**
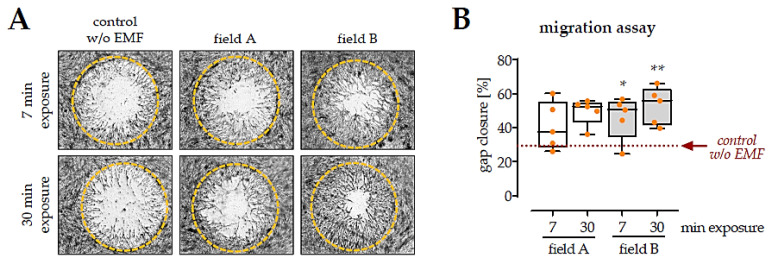
Migration of SCP-1 cells cultured with conditioned medium from macrophages exposed to the ELF-PEMFs. Freshly isolated PBMCs (*n* = 5) were stimulated with 200 nM PMA and exposed to the two ELF-PEMF for 7 min or 30 min, respectively. After 24 h, a conditioned medium was collected and added (1:1) to SCP-1 cells in a migration assay. (**A**) Representative images of the migration assay after 48 h. Cells invading the migration zone were visualized by Sulforhodamine B staining. (**B**) The uncovered area in the migration zone was measured with ImageJ software to quantify the migration assay. Data were compared by non-parametric Kruskal–Wallis test, followed by Dunn’s multiple comparison test: * *p* < 0.05 and ** *p* < 0.01 as compared to control conditions (without ELF-PEMF exposure).

**Figure 6 bioengineering-08-00167-f006:**
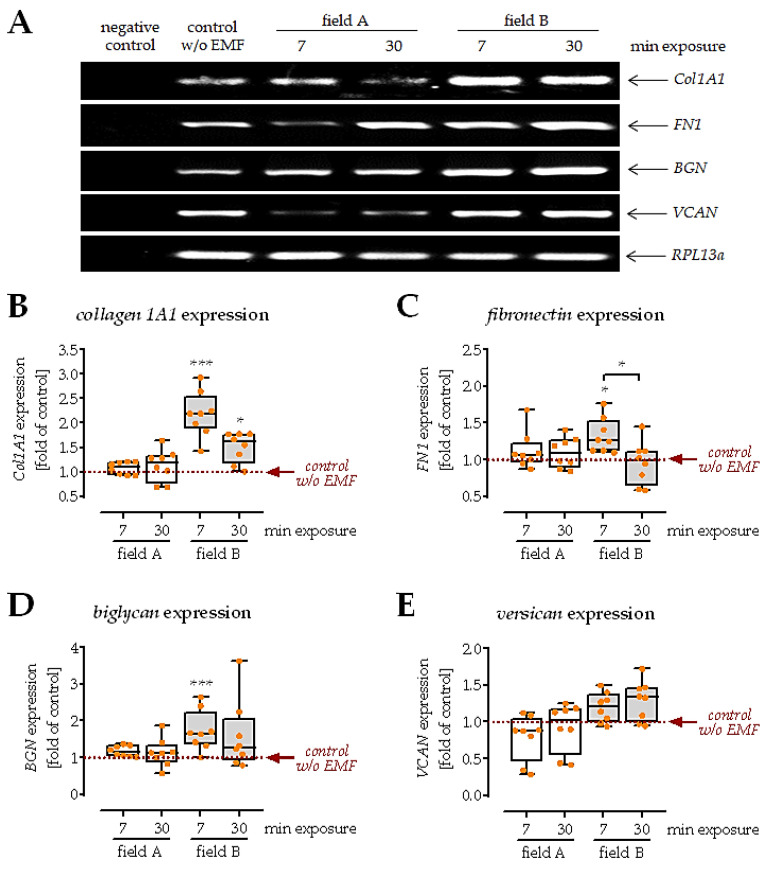
Gene expression of matrix proteins in SCP-1 cells cultured with conditioned medium from macrophages exposed to the ELF-PEMFs for 1 week. Freshly isolated PBMCs (*n* = 5) were stimulated with 200 nM PMA and exposed to the two ELF-PEMF for 7 min or 30 min, respectively. After 24 h, the conditioned medium was collected and added (1:1) to differentiate SCP-1 cells. (**A**) Representative images of the RT-PCR. All uncropped agarose gel images are shown in Appendix A. ImageJ software was used to quantify signal intensities of (**B**) *collagen 1A1* (*Col1A1*); (**C**) *fibronectin* (*FN1*); (**D**) *biglycan* (*BGN*); and (**E**) *versican* (*VCAN*). *RPL13a* was used as a housekeeping gene for normalization. Data were compared by non-parametric Kruskal–Wallis test, followed by Dunn’s multiple comparison test: * *p* < 0.05 and *** *p* < 0.001 as compared to control conditions (without ELF-PEMF exposure).

**Table 1 bioengineering-08-00167-t001:** List of primers, their sequences, and the corresponding PCR conditions.

Target	Gene Bank Accession Number	Sequence Forward Primer	Sequence Reverse Primer	T_a_ [°C]	# of Cycles	Amplicon Size [bp]
RPL13a	NM_012423.3	AAGTACCAGG CAGTGACAG	CCTGTTTCCGT AGCCTCATG	56	30	100
Biglycan	NM_001711.5	CGCCTCGTGT CTCTGCTGGC	GCGGATGCGG TTGTCGTGGA	64	35	501
Versican	NM_001164098.1	AATGCCGTCT GCAGGGTGCC	GGCCGCAAGC GACTGTTCCT	64	35	306
Collagen 1A1	NM_000088.3	CAGCCGCTTC ACCTACAGC	TTTGTATTCAAT CACTGTCTTGCC	56	35	83
Fibronectin	NM_002026.2	CCCCATTCCAG GACACTTCTG	GCCCACGGTA ACAACCTCTT	60	35	203

## Data Availability

The datasets generated during and/or analyzed during the current study are available from the corresponding author on reasonable request.

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
