# Peer review of "Modulation of Macrophage Activity by Pulsed Electromagnetic Fields in the Context of Fracture Healing"

_bioengineering, 2021, doi:10.3390/bioengineering8110167_

Round 1
Reviewer 1 Report
I am very satisfied with the quality of the manuscript. I believe it is fit for publication. Initially I would suggest some adjustments in the characterization of cells, but I understood the objective of the research and will not request any methodological correction. I congratulate the authors.
Author Response
We would like to thank the reviewer for his/her estimate on our manuscript.
Reviewer 2 Report
The authors in this manuscript demonstrates the use of extremely low-frequency pulsed electromagnet fields (ELF-PEMFs) to modulate macrophages. To identify specific ELF-PEMFs that can modulate macrophages, the authors used a wide-range of ELF-PEMFs to assess macrophage attachment and SCP-1 cell migration. After identifying opposing fields "A" and "B", macrophage differentiation and cellular activity were measured by Western, cell-culture, and cytokine array. Authors conclude that ELF-PEMFs can affect macrophages and may be used to modulate immune functions during pathologies such as fracture healing.
The role of immune system in the early stage of fracture healing is crucial, and the study presented in this manuscript provide possible novel avenues that can accelerate fracture healing. However, the following conceptual and technical comments should be addressed to ensure the readers in a similar field can appreciate the effect of ELF-PEMFs.
- Introduction/discussion should address whether ELF-PEMFs can affect both M1 and M2 or only one of the macrophages. During fracture healing macrophages undergo different changes so this would further highlight the effect of ELF-PEMFs.
- Related to the comment above, please indicate the cells that were differentiated and used in this study are M1 or M2 macrophages.
- Authors describe fields "A" and "B" have opposing effects, but according to Fig 1, Field 3 and Field 7 are effect the opposite. Please further clarify/justify the use of fields "A" and "B". It's unclear why having a diverse effects on macrophage function is the reason to exclude field 3.
